# Proangiogenic Effect of 2A-Peptide Based Multicistronic Recombinant Constructs Encoding VEGF and FGF2 Growth Factors

**DOI:** 10.3390/ijms22115922

**Published:** 2021-05-31

**Authors:** Dilara Z. Gatina, Ekaterina E. Garanina, Margarita N. Zhuravleva, Gulnaz E. Synbulatova, Adelya F. Mullakhmetova, Valeriya V. Solovyeva, Andrey P. Kiyasov, Catrin S. Rutland, Albert A. Rizvanov, Ilnur I. Salafutdinov

**Affiliations:** 1Institute of Fundamental Medicine and Biology, Kazan Federal University, 420008 Kazan, Russia; gatina_dilara@mail.ru (D.Z.G.); kathryn.cherenkova@gmail.com (E.E.G.); k.i.t.t.1807@gmail.com (M.N.Z.); gulnazgg12@gmail.com (G.E.S.); mullahmetovaadela@gmail.com (A.F.M.); solovyovavv@gmail.com (V.V.S.); kiassov@mail.ru (A.P.K.); 2School of Veterinary Medicine and Science, University of Nottingham, Nottingham LE12 5RD, UK; Catrin.Rutland@nottingham.ac.uk

**Keywords:** angiogenesis, gene expression, non-viral vectors, 2A-peptides, growth factors, tube formation, VEGF, FGF2, cytokines

## Abstract

Coronary artery disease remains one of the primary healthcare problems due to the high cost of treatment, increased number of patients, poor clinical outcomes, and lack of effective therapy. Though pharmacological and surgical treatments positively affect symptoms and arrest the disease progression, they generally exhibit a limited effect on the disease outcome. The development of alternative therapeutic approaches towards ischemic disease treatment, especially of decompensated forms, is therefore relevant. Therapeutic angiogenesis, stimulated by various cytokines, chemokines, and growth factors, provides the possibility of restoring functional blood flow in ischemic tissues, thereby ensuring the regeneration of the damaged area. In the current study, based on the clinically approved plasmid vector pVax1, multigenic constructs were developed encoding vascular endothelial growth factor (VEGF), fibroblast growth factors (FGF2), and the DsRed fluorescent protein, integrated via picornaviruses’ furin-2A peptide sequences. In vitro experiments demonstrated that genetically modified cells with engineered plasmid constructs expressed the target proteins. Overexpression of VEGF and FGF2 resulted in increased levels of the recombinant proteins. Concomitantly, these did not lead to a significant shift in the general secretory profile of modified HEK293T cells. Simultaneously, the secretome of genetically modified cells showed significant stimulating effects on the formation of capillary-like structures by HUVEC (endothelial cells) in vitro. Our results revealed that when the multicistronic multigene vectors encoding 2A peptide sequences are created, transient transgene co-expression is ensured. The results obtained indicated the mutual synergistic effects of the growth factors VEGF and FGF2 on the proliferation of endothelial cells in vitro. Thus, recombinant multicistronic multigenic constructs might serve as a promising approach for establishing safe and effective systems to treat ischemic diseases.

## 1. Introduction

Currently, ischemic diseases remain one of the leading causes of death in the world’s developed countries [1]. This disease group is characterized by a lack of oxygen and nutrient supply due to impaired micro- and macro-blood supply to a tissue, organ, or limb [2]. Simultaneously, the lack of an adequate blood supply stimulates the activation of angiogenesis processes due to the release of proangiogenic factors [3]. In this regard, a strategy of supportive angiogenic therapy was proposed, which underlies the introduction of exogenous growth factors that promote vasculogenesis and blood circulation restoration [4].

Angiogenesis depends on complex interactions including spatial-temporal interaction of cells and various proangiogenic factors, particularly vascular endothelial growth factor (VEGF) and fibroblast growth factor (FGF2). As the most critical angiogenic factor, vascular endothelial growth factor (VEGF) remains one of the promising candidates for proangiogenic therapy [5,6,7]. The results of numerous in vitro and in vivo experiments have demonstrated that VEGF promotes de novo new blood vessel formation, improves blood flow, and supports myogenesis [8,9]. In addition, binding to the VEGFR1 and VEGFR2 receptors increases vascular permeability and induces proliferation and migration of endothelial cells [10]. It is known that VEGF contributes to the survivability of endothelial cells preventing apoptosis via the PI3K/AKT-signaling pathway and induces expression of the antiapoptotic proteins A1 and Bcl-2 [11,12,13]. The crucial role of VEGF in angiogenic processes has been confirmed in numerous investigations. The knockout of even one allele of the VEGF gene led to embryonic lethality and disorders in the cardiovascular system [14]. VEGF-A inhibitors are widely used in the therapy of solid tumors, thus indicating the significant role of VEGF in both normal and pathological angiogenesis [15]. Over the last decade, several randomized controlled trials have also utilized plasmid and viral vectors with the VEGF gene to treat severe coronary heart disease (Euroinject One, KAT, REVASC, NOTHERN, NOVA, VEGF-A Neupogen, and GENASIS) [6,16]. In 2011, the drug Neovasculgen was approved in Russia (ClinicalTrials.gov identifier: NCT03068585) to treat lower limb ischemia of atherosclerotic genesis, including chronic critical lower limb ischemia. This drug represents a highly purified, supercoiled form of the plasmid pCMV-VEGF165-encoding isoform 165 (a) of vascular endothelial growth factor [17,18].

The basic fibroblast growth factor (FGF2) is the second of the most characterized mitogens utilized in gene therapy protocols to induce therapeutic angiogenesis [19]. FGF2 interacts with the FGFR1 and FGFR2 receptors and with heparan sulfate proteoglycans, which results in proliferation and migration of endothelial cells, protease production, and angiogenesis. Proangiogenic effects of FGF2 have been confirmed in numerous experimental in vivo models of angiogenesis, including the chick embryo chorioallantoic membrane model, the cornea of the eye, and matrigel implants [20]. Moreover, positive effects of FGF2 have been reported in cases of pressure sores, diabetic foot ulcers, and burns. In 2001, Trafermin, a medicine based on human FGF2 for treating pressure ulcers and skin ulcers, was approved in Japan [21]. Likewise, the application of FGF2 has been successful in the treatment of second-degree burns. In the randomized study, patients with deep burns received local injections of either a placebo or bovine FGF2. All patients treated with FGF2 demonstrated faster granulation tissue formation and epidermal regeneration than patients in the placebo group [22]. Similar to VEGF, FGF2 has an affinity for heparin, increases neuronal survival, and reduces apoptotic cell death [23,24]. Unlike VEGF, FGF2 is capable of stimulating the proliferation of not only endothelial cells but also smooth muscle cells and fibroblasts. FGF2 also regulates the recruitment of immune cells such as monocytes, macrophages, and neutrophils, affecting the production of various chemokines [25,26].

The synergetic effects of VEGF and FGF2 were demonstrated in a broad range of studies dedicated to induction of angiogenesis and blood flow restoration [27,28]. For example, it has been described that FGF2 can induce endogenic VEGF expression and its receptors in endothelial cells where VEGFR2 inhibitors had arrested FGF2-induced angiogenesis [29]. Numerous studies have also shown that coordinated activity of VEGF and FGF2 is crucial in all stages of angiogenesis, especially during early embryogenesis, to increase vascular permeability and recruitment of endothelial cells. Stimulation of endothelial cells initiates protease production and plasminogen secretion, destroying basement membrane and activating the invasion of cells in the surrounding matrix [30,31]. Though the exact synergetic mechanism remains unclear, there are several potential signaling pathways involved in both VEGF and FGF2 interactions and interactions with their receptors. Interaction between VEGF–VEGFR is known to activate the Ras-MEK-MAPK and AKT, P38, and PKC signaling pathways, which in turn modulate FGF2 expression and proangiogenic cellular responses [32]. The interaction between FGF2–FGFR also activates Ras-MEK-MAPK signaling pathways and induces VEGF expression [33]. SRC cascade, AKT/P13K, and PKC pathways are known to participate in FGF2-dependent angiogenesis [34]. Thus, interaction between VEGF and FGF2 activate numerous signaling cascades and ultimately stimulate the angiogenic processes in endothelial cells.

Augmented concentrations of proangiogenic factors are known to cause aberrant vessel formation and hemangiomas as well [35]. These results were obtained when viral constructions were used providing longitudinal transgene expression [36]. It is common knowledge that viral vectors represent an efficient vehicle for target gen delivery due to their natural infection capacity of various cell types [37,38]. However, uncontrolled gene expression, immune responses to the constituent components of viral particles, and some viral genomes’ relatively small packaging capacities restrict the feasibility of constructs based on recombinant viruses [39,40]. Compared to viruses, the application of non-viral gene therapy approaches using plasmid vectors do not provoke systemic reactions in the organism, and plasmids could be easily manufactured in preparative amounts [41]. Additionally, the development of new delivery systems has made it possible to achieve high efficiency of transfection both in vitro and in vivo. Continuous improvement of non-viral gene therapy techniques therefore contributes to their widespread use in clinical practice [9].

Currently, several strategies have been proposed to ensure simultaneous gene expression [42,43]: independent internal promoters [44], internal ribosomal entry sites (IRES) [45], mRNA splicing [46], bi-directional promoters, and 2A-peptides [47,48]. However, utilizing several promoters might significantly decrease transcriptional activity and reduce the vector’s packaging capacity. In turn, IRESs provide effective co-expression of a few genes, but their large molecular weight (up to 1 kb) presents a limiting factor [49]. Moreover, the level of transgene expression depends on the location and spatial organization of the transgene. Downstream genes are transcribed less effectively in multicistronic constructions than cistrons located upstream of IRES [50,51,52,53,54].

Picormoviral 2A-sequences represent short peptides, composed of 14-21 amino acid residues, and contain a functional consensus motif Asp-Val/Ile-Glu-Asn-X-Pro-Gly -Pro [55]. This motif interrupts translation, “skipping” the formation of the last glycine-proline bond and releasing an upstream polypeptide that is attached to the C-terminal sequence of 2A. When translating the next polypeptide, proline is used as the first amino acid [56,57]. After 2A-mediated cleavage, the newly synthesized proteins contain a small N-terminal proline and a C-terminal residue of the 2A peptide; however, the proteins usually remain functionally active [57]. Currently, four types of 2A-peptide sequences are utilized in biomedical applications: foot-and-mouth disease virus FMDV 2A (F2A); equine rhinitis virus A (ERAV) 2A (E2A); porcine tospovirus 1 2A (P2A); and asigna virus 2A (T2A) [51]. Mostly, 2A-peptide dedicated studies focus on the efficiency of cleavage of various proteins in multicistronic vectors. For example, it has been demonstrated that T2A usually has the most efficient cleavage compared to other types of 2A-peptides. In contrast, other results have also revealed higher efficiencies of P2A-mediated cleavage in comparison to T2A [51,58]. It is quite important to consider post-translational conformation of target proteins to achieve the most effective transgene expression. For example, the secretion of TGF-β was arrested due to the coding sequence’s upstream position to 2A peptide [59]. Incorporation of the furin cleavage site (Fu) upstream to the 2 sequence enables polyprotein convertase specifically to hydrolyze C-terminal peptide bonds of arginine and cleavage the proteins downstream of R-X-K/R-R motif. This sequence has been discovered in various human proteins, plays a crucial role in protein formation, and enables active secretion and membrane-associated proteins [60,61].

In the present study, using a clinically approved plasmid vector, we designed recombinant constructions containing picornavirus peptide sequences, codon-optimized sequences of VEGF and FGF2, and a reporter DsRed fluorescent protein under the control of a cytomegalovirus promoter. Expression of recombinant proteins, the secretome of modified cells, and the synergetic effects of the target proteins on the formation of capillary-like structures by HUVECs were investigated.

## 2. Materials and Methods

### 2.1. Recombinant Constructs, Isolation, and Restriction Analysis of Plasmid DNA

Based on a clinically approved plasmid vector pVax1 (Thermo Fisher Scientific) [62], multicistronic constructions encoding various gene combinations (pVax1-VEGF-FGF2-DsRed, pVax1-VEGF-DsRed, pVax1-FGF2-DsRed, pVax1-DsRed) were designed and developed (Figure 1). Nucleotide sequences of the VEGF (GeneBank AF486837.1) and FGF2 (GeneBank #DD406196.1) genes were obtained from the NCBI database and cloned under a single CMV promoter. The Fu-cleavage site (AGAAACAGAAGA) and p2A skipping motif (CCACGAAGCAAGCAGGAGATGTTGAAGAAAACCCCGGGCCT) were incorporated between the target genes [63]. To enhance gene expression and reduce restriction sites, an Optimum Gene™ (GenScript, Piscataway, NJ, USA) algorithm was applied. Synthesis of plasmid constructions de novo were carried out using GenScript (https://www.genscript.com/).

### 2.2. Genetic Modification of the Cells with Recombinant Plasmids

Human embryonic kidney cells HEK293T (ATCC CRL-11268) were cultivated on Dulbecco’s modified media supplemented with 10% fetal bovine serum (FBS), 100 U/mL penicillin/streptomycin, and 4 mM L-glutamine. A total of 2.5 × 10^5^ cells were seeded per well in a 24-well plate in two ml medium. Transfections were carried once 70% confluence had been achieved using Turbofect transfection reagent (Thermo Fisher Scientific, Waltham, MA, USA). Transfection efficiency was analyzed after 48 h using DsRed expression by the modified cells as an indicator using a fluorescent inverted micro-scope AxioObserver Z1 Carl Zeiss AG, Oberkochen, Germany).

### 2.3. Quantitative Analysis of mRNA Expression

Total RNA from genetically modified cells was isolated using TRIZOL (Thermo Fisher Scientific, Waltham, MA, USA) according to manufacturer’s instructions. Concentrations of isolated RNA were evaluated using a spectrophotometer NanoDrop 2000 (Thermo Fisher Scientific, Waltham, MA, USA). Complementary DNA (cDNA) was synthesized using a RevertAid kit (Thermo Fisher Scientific, Waltham, MA, USA), also in accordance with the recommended protocol. Expression of target genes was assessed by Real-time PCR and TaqMan technology on a CFX96 Touch Real-Time PCR Detection System (Bio-Rad, Hercules, CA, USA). Non-transfected 293 cells (NTC—non-transfected cells) served as negative controls for quantitative analysis of transgene expression. The relative amount of mRNA for the target genes was normalized to 18S rRNA and calculated using the 2^−ΔΔCT^. Nucleotide sequences of primers specific to codon-optimized sequences of target genes are presented in Table 1.

### 2.4. Immunofluorescent Assays for Transfected HEK293T Cells

Transfected HEK293T cells were primarily fixed with chilled methanol for 20 min at 20°. Cell membranes were permeabilized by adding 0.1% Triton-100 (Helicon, Russia) and then incubated with primary antibodies at a dilution of 1:100, VEGF (Santa Cruze Biotechnology, CA, USA) and FGF2 (Santa Cruze Biotechnology, Santa Cruz, CA, USA), in Tris-buffer saline (TBS) for one hour. Following the primary antibody incubation period, the cells were washed with TBS and coated with the species-specific secondary. Nuclei were stained using a DAPI solution (4′,6-diamidino-2-phenylindole;62248, Thermo Fisher Scientific, Waltham, MA, USA) and photomicrographs for analysis were captured on a fluorescent microscope AxioObserver Z1 (Carl Zeiss AG, Oberkochen, Germany). 

### 2.5. Enzyme-Linked Immunosorbent Assay (ELISA) of Target Genes

The concentrations of VEGF and FGF2 in cell lysates and supernatants of transfected and native cells were evaluated using Human VEGF (DY293B) and Human FGF2 (DY233) kits (R&D Systems, DuoSet, Minneapolis, MN, USA) in accordance with the manufacturers’ recommended protocol. The calibration curve was created based on seven different concentrations in the range of 31.1–2000 pg/mL. Optical densities were measured using BioRadxMark at OD 450. The resulting standard curve was utilized for quantitative analysis of the recombinant proteins.

### 2.6. Multiplex Analysis

Quantitative assessment of secreted soluble factors in conditioned media and cell lysates was measured using a Bio-Plex200 System. based on xMAP Luminex technology, with Bio-Plex Pro Human Cytokine 27-plex Assay (M500KCAF0Y) including the following analytes: IL-1, IL-1ra, IL-2, IL-4, IL-5, IL-6, IL-7, IL-8, IL-9, IL-10, IL-12, IL-13, IL-15, IL-17, FGF2, Eotaxin, G-CSF, GM-CSF, IFN-γ, IP-10, MP-1, MIP-1, MIP-1, PDGF-BB, RANTES, TNF-α, and VEGF. Data collection and analysis was performed using Bio-Plex Manager 4.1 software.

### 2.7. Tube Formation Assay Using Matrigel Matrix

Human Umbilical Vein Endothelial Cells (HUVEC) were isolated and cultivated according to the previously described protocol [43]. Wells in a chilled 96-well plate were coated with 50 µL of Matrigel matrix (with a reduced concentration of growth factors (Becton Dickinson, Franklin Lakes, NJ, USA)) and incubated at 37 °C for an hour. After the matrix had solidified, 1 × 10^4^ endothelial cells were seeded in 100 µL onto the Matrigel surface by adding conditioned medium from the HEK293T cells previously transfected with various plasmid constructs. Non-transfected HEK293T medium and poor medium containing 10% FBS were used as a negative control. Complete medium containing 10% FBS and Endothelial Cell Growth Supplement (ECGS) 30 µg/mL (Sigma-Aldrich, St. Louis, MO, USA) was used as a positive control. The formation of capillary-like structures was assessed after 6 h using phase-contrast microscopy. The total tube length and the number of formed branch nodes were calculated using the WimTube software package in the Wimamsis system (Available online: https://www.wimasis.com/en/WimTube, accessed on 19 March 2019)

### 2.8. Statistical Analysis

Each experiment was performed three times in replicates. All data are presented as mean with standard error of the mean (±s.e.). Multiple comparisons were tested by one-way analysis of variance (ANOVA) followed by a post-hoc Tukey test using the GraphPad Prism 7 software (GraphPad Software, California, USA). The *p*-value *p* < 0.05 was considered statistically significant. Significant probability values were denoted as follows: * *p* < 0.05, ** *p* < 0.01, *** *p* < 0.001, **** *p* < 0.0001, ns—no statistically significant difference.

## 3. Results

### 3.1. Characterization of Multigenic Constructs Containing Picornavirus 2A-Peptide Sequences

In the current study, recombinant plasmid vectors were designed, encoding codon-optimized sequences of VEGF and FGF2 genes and DsRed under the single control CMV promoter. In bi-cistronic (pVax1-VEGF-DsRed, pVax1-FGF2-DsRed) and tri-cistronic vectors (pVax1-VEGF-FGF2-DsRed) the Fu-2A-peptide sequence was incorporated between the target genes. All vectors were constructed based on widely used and clinically approved plasmid pVax1. The primary structure of DNA was verified via routine sequencing (data not shown). The quality of both the purified plasmids, and the presence of target inserts, were confirmed by restriction analysis. The resulting fragments corresponded to the expected molecular size (Figure 1).

**Figure 1 ijms-22-05922-f001:**
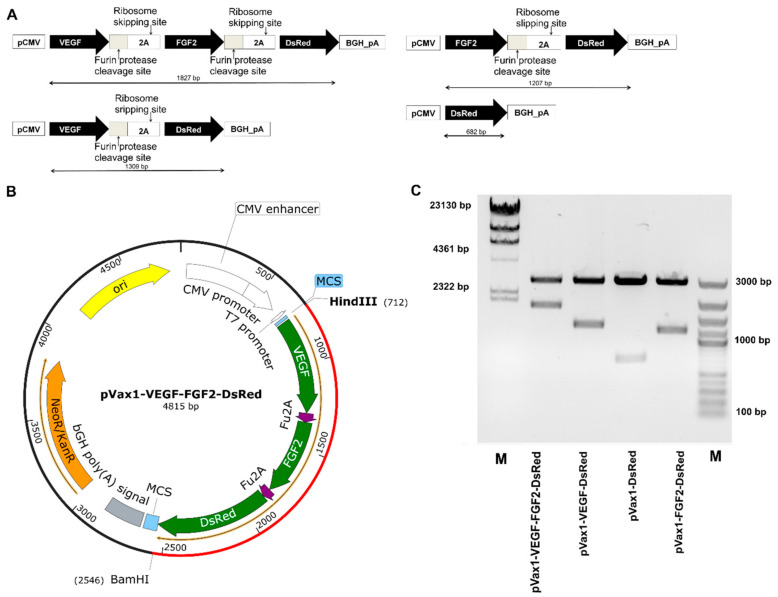
Design and characterization of plasmid vectors. (**A**) Schematic of the genetic cassettes, containing therapeutic genes (Vascular Endothelial Growth Factor (VEGF165), Basic Fibroblast Growth Factor (FGF2)) and reporter fluorescent protein (Red fluorescent protein (DsRed)). (**B**) Schematic diagrams of the expression vectors. (**C**) Analysis of enzymatic digestion. Agarose gel electrophoresis. M—marker/HindIII (SM0103) and GeneRuler DNA LadderMix (SM0331).

### 3.2. VEGF, FGF2 and DsRed Expression in Genetically Modified Cells In Vitro

To confirm the expression of target genes, HEK293T cells were transfected with recombinant plasmid constructs. We found that the expression plasmids increase the expression of mRNA VEGF- and FGF2-modified cells compared with the empty vector (pVax1-Dsred) and non-transfected cells. Fluorescent microscopy analysis revealed DsRed expression in all experimental groups. The immunofluorescent assay demonstrated a positive reaction for antibodies to VEGF and FGF2 in transfected cells (Figure 2A).

### 3.3. Production of VEGF and FGF2 by Genetically Modified Cells

The efficiency of VEGF and FGF2 secretion ex vivo was confirmed using indirect ELISA of cell lysates and supernatants collected from genetically modified cells. The ELISA results revealed statistically significant upregulation of VEGF secretion in both the supernatants and lysates of the cells modified with pVax1-VEGF-FGF2-DsRed (3629.68 ± 125.05 pg/mL). Increased VEGF production was also registered in supernatants of the cells transfected with pVax1-VEGF-DsRed (3530.00 ± 291.15 pg/mL) compared to non-transfected control (61.77 ± 3.03 pg/mL) (Figure 3A). Cells transfected with pVax1-VEGF-FGF2-DsRed (1396.00 ± 29.06 pg/mL) and pVax1-FGF2-DsRed (1728.00 ± 85.18 pg/mL) produced increased amounts of FGF2 compared to the cells modified with pVax1-VEGF-DsRed (16.73 ± 6.09 pg/mL) and in comparison to pVax1-DsRed and naïve cells as well (Figure 3B).

### 3.4. Cytokine Profile Study of Genetically Modified Cells

The effect of the upregulated expression on cytokine pattern was investigated by multiplex analysis of cell lysates and supernatants collected from genetically modified cells. A total of 27 proteins were analyzed and the expression of 15 proteins was detected. Non-transfected cells were shown to express soluble receptor (IL-1ra), pro-inflammatory (IFN-γ, IL-6, IL-7, IL-12(p70),), anti-inflammatory cytokines (IL-8, IL-10), chemokines (IP-10, MCP-1, RANTES), and growth factors (G-CSF, FGF2, GM-CSF, PDGF-bb, VEGF). Secretion levels of several cytokines were insignificantly higher in modified cells. Notably, in cell lysates and supernatants from pVax1-VEGF-FGF2-DsRed group, upregulated IL-1ra, IL-7, IL-8, IL-12(p70), IL-6, IL-10, IFN-γ, IP-10, RANTES, MCP-1, and growth factors G-CSF, GM-CSF, PDGF-bb were observed at levels 3–7 times higher compared to NTC. However, it was statistically not significant (*p* > 0.05). Increased production of the aforementioned factors was also observed in cells transfected with pVax1-VEGF-DsRed. Slight augmentation of IL-6, IL-8, MCP-1, IFN-γ, and PDGF-bb were registered in cell lysates in the pVax1-FGF2-DsRed group (*p* > 0.05). An expected augmentation of VEGF and FGF2 (*p* < 0.01) was scored in pVax1-VEGF-DsRed, pVax1-FGF2-DsRed, and pVax1-VEGF-FGF2-DsRed transfected cells (Figure 4) that coincided with the above-mentioned ELISA data (Figure 3).

### 3.5. Effect of Increased VEGF and FGF2 Expression on Capillary-Like Structures Formation by HUVEC In Vitro

To evaluate the proangiogenic potency of plasmid constructs and the synergetic effect of secreted VEGF and FGF2, HUVEC cells were cultured on a three-dimensional Matrigel matrix supplemented with conditioned media from transfected cells (Figure 5). Conditioned media collected from cells transfected with pVax1-VEGF-FGF2-DsRed provoked the most significant stimulation on tube formation by HUVEC compared with cells supplemented with media obtained from pVax1-VEGF-DsRed, pVax1-FGF2-DsRed, pVax1-DsRed, and control HEK293T cells. Meanwhile, in the control groups (pVax1-DsRed and NTC), a few cellular structures that were not connected were observed. In contrast, other groups demonstrated a higher level of organization of capillary-like structures (tubule area and total length) and a higher level of tubular organization (number of loops and bifurcations). In the pVax1-VEGF-DsRed group the total length of formed tubes was less (166640.30 ± 1297.53px) compared to pVax1-VEGF-FGF2-DsRed (21409.00 ± 2183.81px). However, in the pVax1-FGF2-DsRed group (20150.30 ± 1289.05px), the total length of the formed tubes was not significantly different from the pVax1-VEGF-FGF2-DsRed group21409.00 ± 2183.81px) (Figure 5B). Similar results were obtained concerning the number of branch points discovered; in the pVax1-VEGF-FGF2-DsRed (190 ± 28.86), pVax1-VEGF-DsRed (139.00 ± 15.30), and pVax1-FGF2-DsRed (187.66 ± 16.259), the numbers were higher in comparison to the control groups pVax1-DsRed (86.66 ±15.45) and NTC (70.00 ± 3.7) (Figure 5C). Thus, augmented expression of VEGF and FGF2 contributed to the formation of a denser network of capillary structures compared to other groups, resulting in increased amount of loops forms, with pVax1-VEGF-FGF2-DsRed (71.5 ± 11.61), pVax1-VEGF-DsRed (44.25 ± 6.00), and pVax1-FGF2-DsRed (69.66 ± 11002) showing higher numbers than control groups pVax1-DsRed (20.33 ± 7.12) and NTC (11.33 ± 2.18) (Figure 5D). In the pVax1-VEGF-FGF2-DsRed and pVax1-FGF2-DsRed groups, the percentage of cell coverage increased by 10% compared to control groups pVax1-DsRed and NTC. This up-regulation was statistically significant and correlated with other indexes, including the differential number of loops, branch points, and total tube length (Figure 5E).

## 4. Discussion

We designed and tested multigene vectors encoding the FGF2, VEGF, and DsRed genes, or combinations of these, combined through the picornoviral 2A-peptide sequences. Our data demonstrated that the expression of mRNA and FGF2 and VEGF proteins in vitro was higher than that observed in the control groups (cells modified with the DsRed plasmid and non-modified cells).

The selection of an appropriate vector system is believed to be one of the essential aspects of developing gene therapy drugs. The vector pVax1 is a non-immunogenic optimized expression system. The content of eukaryotic DNA sequences is critically reduced, thus minimizing the vector’s chromosomal integration into the host cell genome. Besides this, pVax1 has been approved by the American Food and Drug Administration (FDA) and is currently used widely to develop DNA vaccines [64]. Considering the world’s current situation due to the global spread of severe acute respiratory syndrome 2 (SARS-CoV-2), DNA vaccine development remains an urgent issue [65]. In particular, multicistronic recombinant constructs could be used for the simultaneous equimolar expression of several immunogenic proteins of differing pathogens.

Gene co-expression systems make it feasible to increase the induction efficiency of therapeutic angiogenesis due to the simultaneous delivery of several proangiogenic factors. Application of 2A-peptides may become a strategy employed to achieve stable gene co-expression [66]. This approach is attractive in the expression systems’ design when encoding combinations of therapeutic molecules in the same transcriptional unit and provides a large packaging capacity [67]. In addition, it has been tested in a wide range of eukaryotic cells [68,69]. High transgene expression has been confirmed, as has good cleavage of synthesized recombinant proteins [50,68], and the absence of immune responses to system components has also been verified [68]. The application of multicistronic expression systems will therefore simplify the development procedures needed for polygenic constructs to achieve the simultaneous delivery of several therapeutic genes to a particular cell [70,71]. In the current study, HEK293T cells were used as a model system to characterize recombinant plasmids and evaluate the effect of over-expression of VEGF and/or FGF2 on the secretion profile of genetically modified cells. The possibility of successful modulation of the secretion profile of HEK293 by the action of various external factors has been demonstrated previously. In particular, co-cultivation of HEK293 with mesenchymal stem cells that expressed lipocalin 2 resulted in an increased expression of growth factors HGF, IGF-1, and FGF2 by HEK293 cells [72]. Confirmed changes in the secretion of HEK293 IL-2, IL-4, IL-6, IL-8, IL-10, GM-CSF, IFN-γ, TNF-α due to the influence of bovine serum albumin have also been published [73].

In our study, multiplex analysis of soluble factors in the supernatants and cell lysates of HEK293T revealed augmentation of several cytokines, chemokines, and growth factors in transfected cells. However, the differences found, except for VEGF and FGF2, were not statistically significant. Therefore, we suggest that for a deeper understanding of the functioning and effects of VEGF and FGF2 overexpression on the state of modified cells, it is necessary to expand the range of analytes investigated, as well as to select cells that have a pronounced autocrine and paracrine potential relating to the production of the recombinant factors used. In our opinion, mesenchymal stem cells from various origins could serve as appropriate candidates. In particular, we have previously demonstrated that genetic modification and overexpression of proangiogenic factors by cells can modulate the secretory profile of stem cells from human adipose tissue [74]. Shanshan Jin et al., previously reported that overexpression of FGF2 by human gingival mesenchymal stem cells enhanced their secretion of VEGF and TNF-β [75]. Similar results were obtained by Yukita et al., where recombinant FGF2 increased the expression of glial neurotrophic growth factor (GDNF) by dental follicle cells [76]. We assume that such adaptations of the secretome to overexpress angiogenic factors might further increase the transplanted cells’ therapeutic potential. The current assay has shown a high correlation between the various techniques used to verify the created structures’ functionality. Using an in vitro model of angiogenesis, we have shown that the conditioned medium collected from genetically modified cells caused a stimulating effect on capillary formation by endothelial cells. The presented data are consistent with our earlier obtained results studying two-cassette plasmid vectors providing independent expression of two growth factors [43]. The results indicate that, in the studied model, FGF2 more effectively affects the formation of capillary-like structures by human endothelial cells as compared to VEGF. The slight effect of secreted VEGF on the formation of vascular-like structures in the present study is likely to be due to the short exposure period. When using recombinant VEGF or VEGF expressing plasmid constructs, several studies have demonstrated that significant induction of angiogenesis is only observed 20–30 days after therapeutic exposure [77,78]. At the same time, significant differences were not observed between the pVax1-VEGF-FGF2-DsRed and pVax1-FGF2 groups in the present study. We assume that the critical contribution to the formation of vascular-like structures in vitro is mediated by FGF2 expression. These results are consistent with previous reports showing that FGF2 induced the highest blood vessels density in a mouse cornea model [79]. Cartland et al. have previously indicated that FGF2 is more effective at stimulating angiogenesis in vitro and in vivo than VEGF and TRAIL [80]. This phenomenon may be because VEGF and FGF2 act on a wide range of receptors and, accordingly, activate various signaling cascades and stages of angiogenesis [79,80,81].

It is worth emphasizing that pro-angiogenic factors are active even in picomolar concentrations and, as a consequence, a slight local increase in their concentration is sufficient to achieve a therapeutic effect. Moreover, in different living species, presumably, since trophic factors are evolutionarily conserved molecules, the same factors can activate similar biological effects. In this regard, it is therefore no coincidence that human molecules, such as growth factors, might be active in various model organisms. Moreover, codon optimization is expected to preserve the biological potential of the synthesized molecules [82]. Simultaneously, alternative data are accumulating, and indicating the negative effect of codon optimization on the translation and structure of the synthesized protein [83], which introduces one more variable that must be taken into account when creating effective, optimized gene therapy systems. Our results show that recombinant constructs expressing different genes are putatively implementing discrete patterns of angiogenesis. Their efficacy for therapeutic use in the induction of angiogenesis should be tested using in vivo models.

## 5. Conclusions

Our work has constructed and tested, in vitro, a multicistronic system that ensures simultaneous efficient delivery of several therapeutic growth factor genes into cells. Data on the functionality of the synthesized recombinant proteins as the part of the multicistronic construct have also been obtained. We have demonstrated that the proangiogenic factors synthesized in the secretome of genetically modified cells exhibit stimulating effects on the formation of capillary-like structures by HUVEC in vitro. This approach can be used to develop and test gene therapy protocols for various human diseases that require the simultaneous expression of several transgenes to stimulate therapeutic angiogenesis and other regenerative processes.

## Figures and Tables

**Figure 2 ijms-22-05922-f002:**
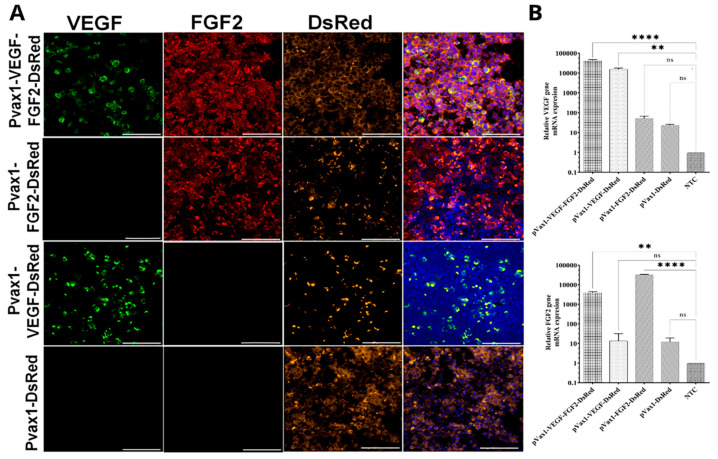
Expression of recombinant angiogenic factors in transfected HEK293T cells. (**A**) Immunofluorescent analysis of genetically modified HEK293T cells. Staining for VEGF (green) and FGF2 (red). Nuclei were counterstained using a DAPI solution (4′,6-diamidino-2-phenylindole) (blue). Scale bar 100 µm. (**B**) mRNA expression of target genes (VEGF, FGF2) in HEK293T cells transfected with plasmids pVax1-VEGF-FGF2-Dsred, pVax1-VEGF-Dsred, pVax1-FGF2-Dsred, pVax1-Dsred. mRNA from cells was assayed by RT–PCR and quantified relative to 18S rRNA mRNA levels. mRNA expression in non-transfected cells (NTC) was considered as control. Data presented as average ± s.e.; *p* < 0.05 * regarded as statistical significant differences (*n* = 3; ** *p* < 0.01; **** *p* < 0.0001 compared with control; ns—non-significant).

**Figure 3 ijms-22-05922-f003:**
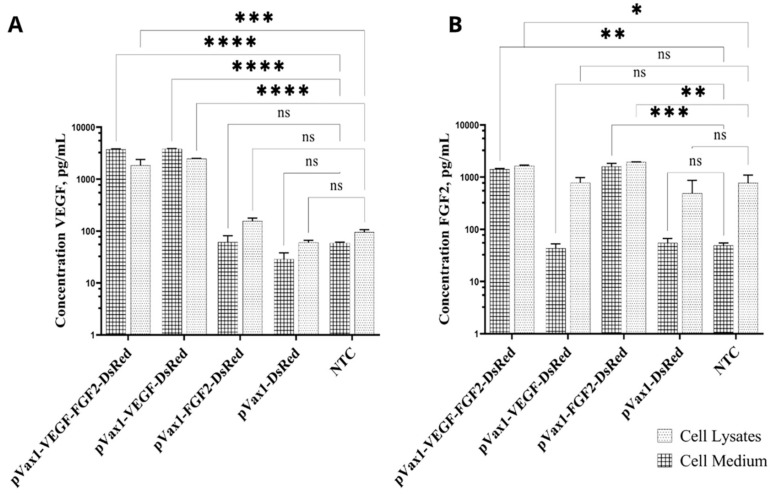
Analysis of VEGF and FGF2 concentrations in HEK293T cells transfected with obtained plasmid constructions. A total of 2.5 × 10^5^ were seeded per well in a 12-well plate, and transfections were conducted after letting the cells adhere overnight. (**A**) VEGF concentration in supernatants and cell lysates of modified cells (pVax1-VEGF-FGF2-DsRed, pVax1-VEGF-DsRed, pVax1- FGF2-DsRed, pVax1-DsRed) and non-transfected controls (NTC); (**B**) FGF2 concentration in supernatants and cell lysates of modified cells (pVax1-VEGF-FGF2-DsRed, pVax1-VEGF-DsRed, pVax1- FGF2-DsRed, pVax1-DsRed) and non-transfected controls (NTC). Data presented as average ± s.e. of three independent repeats in independent samples, statistically significant differences * *p* < 0.05; ** *p* < 0.01; *** *p* < 0.0001, **** *p* < 0.0001; ns—non-significant.

**Figure 4 ijms-22-05922-f004:**
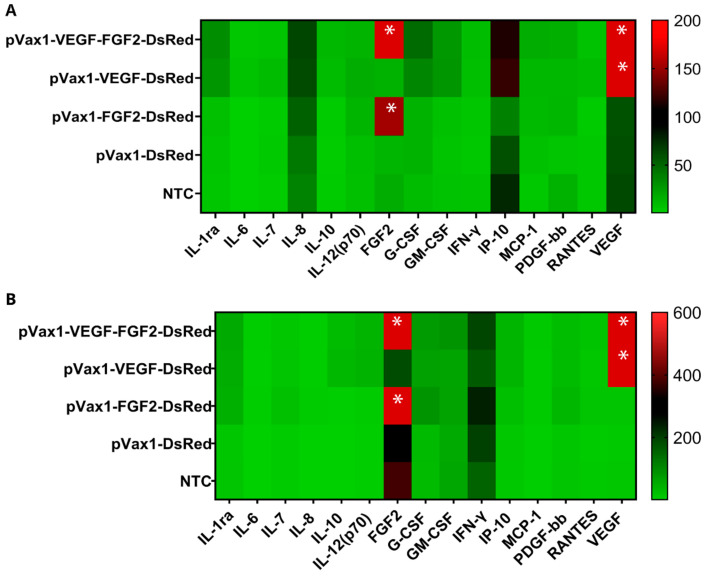
Protein array analysis of cytokine, chemokine, and growth factor in genetically modified HEK293T cells (pg/mL). (**A**) Concentration of soluble molecules, secreted by transfected HEK293T cells. (**B**) Concentrations produced from factors in lysates from transfected HEK293T cells. Analytes was measured 48 h after cell transfection. Data presented as average (n = 4), asterisks (*) indicate statistical significance compared with the control, as determined by one-way analysis of variance (ANOVA) followed by a post-hoc Tukey test (** *p* < 0.01).

**Figure 5 ijms-22-05922-f005:**
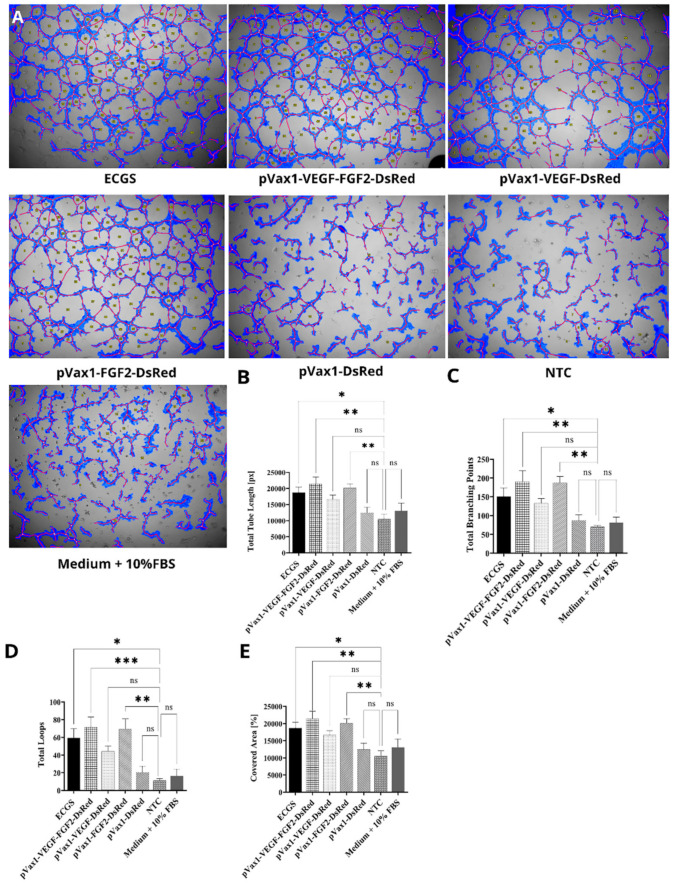
Analysis of tube formation by human umbilical vein endothelial cells (HUVEC). A total of 1 × 10^4^ HUVEC were seeded per well in a 96-well plate. HUVECs were cultivated with conditioned media collected from HEK293T cells transfected with pVax1-VEGF-FGF2-DsRed, pVax1-VEGF-DsRed, pVax1-FGF2-DsRed, pVax1-DsRed, and non-transfected control as well (NTC). ECGS—complete medium containing 10% FBS and Endothelial Cell Growth Supplement (30 µg/mL). Medium+ 10 FBS –medium containing 10% FBS without ECGS. (**A**) Brightfield microscopy image, scale bar 200 µm. (**B**) Length of formed tubes (measured in px), (**C**) Number of branch points formed, (**D**) Number of loops, (**E**) Area of cell coverage. Analysis was carried using Wimasis imaging software package. Data presented as average ± s.e. * indicates statistically significant data relative to control (n = 3; ** *p* < 0.01; *** *p* < 0.0001 ns—non-significant).

**Table 1 ijms-22-05922-t001:** Nucleotide sequences of primers and probes used for qPCR.

Primer Name	Nucleotide Sequence
rmh-18s-TMF	GCCGCTAGAGGTGAAATTCTTG
rmh-18s- TMR	CATTCTTGGCAAATGGTTTCG
rmh-18s-prode	(HEK)-ACCGGCGCAAGACGGACCA-(BHQ)
co-VEGF165-F	CAGATCATGGGGATCAAGCC
co-VEGF165-R	CATGGATTCTCCTGCCTTGC
co-VEGF165-Probe	(6-FAM)-CCAGGGCCAGCACATCGGCG -(BHQ1)
co-FGF2-F	GAGGCTGTACTGCAAGAACG
co-FGF2-R	TGATAGACACCAACGCCTCTC
co-FGF2-R	(6-FAM)-CCTCGGCCTGCAGCTGCTGCAGC -(BHQ1)

## Data Availability

The data presented in this study are available on request from the corresponding author. The data are not publicly available due to privacy.

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
