# Peer review of "Proangiogenic Effect of 2A-Peptide Based Multicistronic Recombinant Constructs Encoding VEGF and FGF2 Growth Factors"

_ijms, 2021, doi:10.3390/ijms22115922_

Round 1
Reviewer 1 Report
In the study, the authors designed a recombinant construction containing picornavirus peptide sequences, codon-optimized sequences of VEGF, FGF2 and a reporter DsRed fluorescent protein under the control of a cytomegalovirus promoter. Expression of recombinant proteins of VEGF and FGF2 in transfected cells were significantly higher than control cells, and the secretome of modified cells enhanced the formation of capillary like structures by HUVECs. The paper was well written, but the results were more like preliminary. Here are a few my suggestions and questions:
1: Fig 2: Staining for VEGF (green) and FGF2 (red), were the images for pVax1-VEGF-Dsred and pVax1-FGF2-Dsred swapped?
2: Fig 4: What does the white color bar mean, highest level? Need more details in the figure legend.
3: Fig 5: Suggest the authors transfect HUVEC or mesenchymal stem cells to see if similar results would be obtained.
4: Suggest the authors use matrigel plug containing transfected endothelial cells in an animal study to further confirm the system works in vivo.
Author Response
Dear reviewer
Thank you for your helpful comments and for taking the time to point out options to improve our manuscript. We have revised the manuscript following both reviewers' suggestions.
1: Fig 2: Staining for VEGF (green) and FGF2 (red), were the images for pVax1-VEGF-Dsred and pVax1-FGF2-Dsred swapped?
You are absolutely right. We have accordingly made the necessary changes to the manuscript.
2: Fig 4: What does the white color bar mean, highest level? Need more details in the figure legend.
We have accordingly made the necessary changes to the manuscript.
3: Fig 5: Suggest the authors transfect HUVEC or mesenchymal stem cells to see if similar results would be obtained.
In the future, we plan to conduct this study using HUVEC or mesenchymal stem cells. It should also be emphasized that the transfection of primary cells with a greater response to exogenous and endogenous factors was carried out by us earlier. We have shown that genetic modification of human adipose-derived stem cells (ADSC) with recombinant plasmids encoding therapeutic growth factors (VEGF and FGF2) changed the therapeutic potential of these cells. We reported that genetic modification of human ADSC with recombinant plasmid results in an expected increase of VEGF secretion and also results in the increase of IL-8 and MCP-1 secretion into the culture medium (https://www.researchgate.net/publication/288422821_Genetic_modification_of_adipose_derived_stem_cells_with_recombinant_plasmid_DNA_pBud-VEGF-FGF2_results_in_increased_of_IL-8_and_MCP-1_secretion). In this case, we can get more hypothesized that we will see fluctuations in the secretion profile of specific proteins associated with VEGF and FGF2 expression.
4: Suggest the authors use matrigel plug containing transfected endothelial cells in an animal study to further confirm the system works in vivo.
In this study, we have concentrated on in vitro characterization of generated recombinant constructs. Therefore, the study of the matrigel plug containing transfected cells was outside the scope of this study. However, at present, we have received preliminary data on the induction of angiogenesis by the recombinant constructs presented in this study with the introduction of a matrigel plug containing transfected cells on Nude mice. We were shown that isolated Matrigel plugs with genetically modified cells contain a greater amount of haemoglobin and have an increased level of mRNA expression (VEGF and FGF2). In addition, cells express recombinant proteins, secrete various cytokines, chemokines and growth factors, and have an increased content of mRNA (CD31, VE-cad, Velerbrand factors, etc.) (Meeting Abstract Recombinant plasmids containing picornaviral self-cleaving 2A-peptides and expressing VEGF and FGF2 growth factors induce angiogenesis in vivo ESGCT 27th Annual Congress in collaboration with SETGyc Meeting HUMAN GENE THERAPY Vol: 30 N:11 P.: A200-A200)
Reviewer 2 Report
The manuscript titled “Proangiogenic effect of 2A-peptide based multicistronic recombinant...” authored by Dilara Z. Gatina et al. demonstrated that the multicistronic multigene vectors consistent with VEGF and FGF2 genes enhances HUVEC tube formation without significant shift in the general secretory cytokines, to utilized forced co-expression to co-stimulate the signaling cascade, proposing a unique therapeutic approach. However, there are several points that need clarification in regards to the design of the study and results presentation.
[Major points]
Authors designed and created multicistronic expression vectors for a new therapeutic approach. Fu-2A-peptide-sequence was used to multicistronic expression, but there is no sequence information on Fu-cleavage sites and p2A peptide-sequence used in this study in the Materials and Methods section. Authors may address what is the novel sequence used in the manuscript whereas there many commercially available p2A multicisronic vectors. Authors also can be addressed the advantage of their newly designed vector system compare to other co-expression system (such as dual expression vector pBud-VEGF165-VEGF2, which author's recent published paper in Cells 2021, also enhances angiogenesis) or conventional medication for ischemic diseases.
The texts in figures in Figure 1 are hard to read. Please increase the image resolution or use vector graphics. The cleaved VEGF and FGF2 may confirmed their molecular size by western blot.
Figure 2 is problematic. Vector transfected cells were subjected to the immunostaining of VEGF and FGF2 antibodies. The pVax1-VEGF-DsRed transfected cells did not stained with VEGF antibody but stained with FGF2, whereas the mRNA expression of VEGF was increased in B. The similar problems presented in pVax1-FGF2-DsRed vector transfected cells, too. The FGF2 expression did not observed in immunostaining but FGF2 mRNA expression was increased. These inconsistences should be explained. The immunostaining in blue color did not indicate in figure legend in Figure 2. Again, the texts in figure are hard to view, especially the statistical significance marks are indistinguishable. Please increase the resolution or use vector graphics.
The authors address concentration of VEGF and FGF2 in the conditioned medium from vector transfected 293 cells in Figure 3 without internal control. How many cells were used for each CM samples?
Please describe the meaning of *** in Figure legend in Figure 3.
The authors point there is no gene expression shift on their secretomes in Figure 4. Authors describes “Secretion levels of several cytokines were insignificantly higher in modified cells” and "these did not lead to a significant shift on the general secretory profile...". Significance should statistically evaluate. How many experiments were performed? Data points indicates mean? Authors may address on this. Please indicate the reason comparing cytokines in medium (A) and in lysate (B) in Figure 4.
In Figure 5, I don’t convince how the conditioned medium enhances the tube formation of HUVEC. HUVEC with Non-transfected control (NTC) looks inhibited tube formation, because they don’t have positive control for the tube formation assay. Authors may include positive controls of HUVEC with or without purified VEGF, FGF2, VEGF+FGF2 to show the proper tube formation in the same time point.
[Minor points]
Keywords in first page is empty.
Line 51, “migra-tion” should “migration”
In Materials and Methods, line 159, please indicates the Accession Number of VEGF and FGF2 sequences they obtained from NCBI database.
Line 214, “50 l of...” and line 216 “100 l onto...”, please check the text.
Lines 224-225, in the Statistical analysis section, “Each experiment was repeated at least three times independently using cells obtained from different donors.”, does it indicate only for figure 5 experiment using HUVEC? Please indicate the methods for statistical analysis in other experiments, too.
In figure legend in Figure 2, there is no indication of “A” and “C”.
In the manuscript, “Genetically-modified cells” and “stable transgene co-expression” for the description of vector transfected 293 cells, but transfection was performed as transient transfection (24-48h post transfection). pVax1 vector designed as minimized genome integration for vaccine and primarily it may not occur genome integration. Please reconsider the description.
Author Response
Dear reviewer
Thanks for your critical reading of our manuscript and important remarks. We added some recommended corrections. In the present study, we used the p2A sequence early described Garanina E.E. et al. 2015 (https://doi.org/10.1038/sc.2015.162). We added the sequences were used in the current manuscript.
The texts in figures in Figure 1 are hard to read. Please increase the image resolution or use vector graphics. The cleaved VEGF and FGF2 may confirmed their molecular size by western blot.
We changed Figure 1 and increased the image resolution. Because we are using the previously described p2A system, here we have verified the expression of only proteins by ICC and ELISA. The cleaved VEGF and FGF2 were confirmed by our group before (Garanina E.E. et al. 2015 (https://doi.org/10.1038/sc.2015.162)).
Figure 2 is problematic. Vector transfected cells were subjected to the immunostaining of VEGF and FGF2 antibodies. The pVax1-VEGF-DsRed transfected cells did not stained with VEGF antibody but stained with FGF2, whereas the mRNA expression of VEGF was increased in B. The similar problems presented in pVax1-FGF2-DsRed vector transfected cells, too. The FGF2 expression did not observed in immunostaining, but FGF2 mRNA expression was increased. These inconsistences should be explained. The immunostaining in blue color did not indicate in figure legend in Figure 2. Again, the texts in figure are hard to view, especially the statistical significance marks are indistinguishable. Please increase the resolution or use vector graphics.
Figure 2 shows the legends incorrectly. We changed them. We corrected and increased the resolution. The expression did not observe in immunostaining, but FGF2 mRNA expression was increased. We think this due to higher sensitivity, PCR compared to the immunological method - immunocytochemistry. Тherefore we were not able to recognise the expression of FGF2 and VEGF in unmodified cells using ICC.
The authors address concentration of VEGF and FGF2 in the conditioned medium from vector transfected 293 cells in Figure 3 without internal control. How many cells were used for each CM samples?
Please describe the meaning of *** in Figure legend in Figure 3.
We have accordingly made the necessary changes to the manuscript. In all our experiments, HEK293 cells transfected pVax1-DsRed can be seen as the internal control because they are corresponding empty vector without therapeutic genes.
The authors point there is no gene expression shift on their secretomes in Figure 4. Authors describes “Secretion levels of several cytokines were insignificantly higher in modified cells” and "these did not lead to a significant shift on the general secretory profile...". Significance should statistically evaluate. How many experiments were performed? Data points indicates mean? Authors may address on this. Please indicate the reason comparing cytokines in medium (A) and in lysate (B) in Figure 4.
We have accordingly made the necessary changes to the manuscript. Evaluation of the cytokine representation in the medium and cell lysate was necessary to confirm the biosynthesis of the examined factors by cells rather than their presence in the growing medium
In Figure 5, I don’t convince how the conditioned medium enhances the tube formation of HUVEC. HUVEC with Non-transfected control (NTC) looks inhibited tube formation, because they don’t have positive control for the tube formation assay. Authors may include positive controls of HUVEC with or without purified VEGF, FGF2, VEGF+FGF2 to show the proper tube formation in the same time point.
Thank you very much for your comments and suggestion. We agree that inhibition may seem to occur. However, the purpose of the research project is not to evaluate the effect of recombinant proteins but to verify the recombinant constructs. In this regard, the comparison with control and internal control () seems more correct to us. At the same time, we agree with your critique. In this regard, as a positive control, we added a previously not included group with ECGs (Endothelial Cell Growth Supplement). Of course, ECGS is not a purified VEGF and/or FGF2, but it can also be used as a positive control. (DOI: 10.1186/s13287-018-0815-3) (Endothelial Cell Growth Supplement (Containing optimized concentrations of: Human FGF basic, Human LR3 IGF-1, Human VEGF165, Human EGF, Heparin, Hydrocortisone, L-Ascorbic Acid, Fetal Bovine Serum).
In the manuscript, “Genetically-modified cells” and “stable transgene co-expression” for the description of vector transfected 293 cells, but transfection was performed as transient transfection (24-48h post transfection). pVax1 vector designed as minimized genome integration for vaccine and primarily it may not occur genome integration. Please reconsider the description.
There is indeed a discrepancy between the terms. Therefore We slightly reconsider the description. In the present manuscript, we demonstrate the transient transfection of HEK293 cells. Nevertheless, the cells have been genetically modified. In this regard, we determined to leave the term "Genetically-modified cells." Simile term present in articles https://doi.org/10.1186/1472-6750-7-90 and https://www.nature.com/articles/gt200980
Round 2
Reviewer 1 Report
I have a few questions regarding Fig 4:
1: Why are there 2 As and 2Bs? It is very confusing. They look similar. Please keep one (the bottom A and B?) .
2: What do those white color bar indicate in top A and B ? The authors did not address my comment in the first review. If you remove these figures, you dont have to answer.
3: The results in Fig 4 and description in the results are not consistent. VEGF and FGF were the only ones statistically significantly changed. However, the authors did described a lot of other targets such as cytokine and chemokines also changed, but no statistical significance was observed (mentioned only in discussion), Please remove them or state clearly that there are no statistical significance in the results section, not just in discussion.
Author Response
Dear reviewer
Thanks for your critical reading of our manuscript and new vital remarks.
We added some recommended improvements.
I have a few questions regarding Fig 4:
1: Why are there 2 As and 2Bs? It is very confusing. They look similar. Please keep one (the bottom A and B?) .
Early we changed figure 4 and added a new figure in the manuscript. The upper figure deleted from the final version of the manuscript. Figures A and B are presented various conditions. For example, Fig. A displayed concentration cytokines in supernatants, and Fig. Ð’ showed concentration cytokines in cell lysates.
2: What do those white color bar indicate in top A and B ? The authors did not address my comment in the first review. If you remove these figures, you dont have to answer.
We addressed your comment in the first review. Therefore, white colour bars have removed in the latest version of the manuscript.
3: The results in Fig 4 and description in the results are not consistent. VEGF and FGF were the only ones statistically significantly changed. However, the authors did described a lot of other targets such as cytokine and chemokines also changed, but no statistical significance was observed (mentioned only in discussion), Please remove them or state clearly that there are no statistical significance in the results section, not just in discussion.
You are absolutely right. Only VEGF and FGF were statistically significantly changed in the comparison groups. In the text of the manuscript, we added statistical significance values (statistically not significant (p > 0.05); statistically significant (p < 0.01)) and marked (asterisk) statistically significant data in figure 4.
Reviewer 2 Report
The authors vastly improved their manuscript from the comments and suggestions. I agree with the author's response and corrections and recommend to accept for publication.
Author Response
Dear reviewer.
Thanks for your reading of our manuscript. We think the manuscript became better after your editing. Thank you for your work.
Round 3
Reviewer 1 Report
No more comments